# The *Reeler* Mouse: A Translational Model of Human Neurological Conditions, or Simply a Good Tool for Better Understanding Neurodevelopment?

**DOI:** 10.3390/jcm8122088

**Published:** 2019-12-01

**Authors:** Laura Lossi, Claudia Castagna, Alberto Granato, Adalberto Merighi

**Affiliations:** 1Department of Veterinary Sciences, University of Turin, I-10095 Grugliasco (TO), Italy; laura.lossi@unito.it (L.L.); claudia.castagna@unito.it (C.C.); 2Department of Psychology, Catholic University of the Sacred Heart, I-20123 Milano (MI), Italy

**Keywords:** reelin, LIS2, ADLTE, autism, schizophrenia, translational models, GABAergic interneurons, dendritic spines, forebrain, cerebellum

## Abstract

The first description of the *Reeler* mutation in mouse dates to more than fifty years ago, and later, its causative gene (*reln*) was discovered in mouse, and its human orthologue (*RELN*) was demonstrated to be causative of lissencephaly 2 (LIS2) and about 20% of the cases of autosomal-dominant lateral temporal epilepsy (ADLTE). In both human and mice, the gene encodes for a glycoprotein referred to as reelin (Reln) that plays a primary function in neuronal migration during development and synaptic stabilization in adulthood. Besides LIS2 and ADLTE, *RELN* and/or other genes coding for the proteins of the Reln intracellular cascade have been associated substantially to other conditions such as spinocerebellar ataxia type 7 and 37, *VLDLR*-associated cerebellar hypoplasia, *PAFAH1B1*-associated lissencephaly, autism, and schizophrenia. According to their modalities of inheritances and with significant differences among each other, these neuropsychiatric disorders can be modeled in the homozygous (*reln*^−/−^) or heterozygous (*reln*^+/−^) *Reeler* mouse. The worth of these mice as translational models is discussed, with focus on their construct and face validity. Description of face validity, i.e., the resemblance of phenotypes between the two species, centers onto the histological, neurochemical, and functional observations in the cerebral cortex, hippocampus, and cerebellum of *Reeler* mice and their human counterparts.

## 1. Introduction

Neuronal migration and precise setting during neurogenesis depend, among others, on reelin (Reln), a 388 kDa glycoprotein secreted by certain neurons within the extracellular matrix [1,2]. The name was given to the protein after the detection of its coding gene, and the acknowledgement that its lack was causative of the mouse *Reeler* mutation [3], which was described, about half a century before, consisting in a form of ataxia [4]. The mutation is autosomic and shows recessive transmission. Consequently, only homozygous recessive *Reeler* mice (*reln*^−/−^) are totally devoid of Reln and have a definite phenotype. Behaviorally, the latter consists of dystonia, ataxia, and tremor; structurally it primarily affects the design of the cerebral cortex, hippocampus, and cerebellum [5,6]. Contrarily to the mutants, the phenotype of heterozygous *Reeler* mice (*reln*^+/−^) is normal, but, interestingly, these animals may be translational models of certain human neuropsychiatric disorders [7].

Shortly after the original discovery, it became clear that the mouse gene (*reln*) had a very high homology to that in humans (*RELN*) [8]. Then, a few years later, it was shown that autosomic recessive mutations of the *RELN* gene were linked to a form of lissencephaly with cerebellar hypoplasia (LCH) [9], with associated findings suggested that *RELN* was linked to some neuropsychiatric conditions [10], and *RELN* was demonstrated to be reduced in the cerebellum of autistic patients after Western blotting and immunodetection [11].

Determining a good translational mouse model for a neuropsychiatric condition needs construct, predictive, and face validity [12]. Rigorously, construct validity only relates to transgenic mice, but, in a broader definition, it also comprehends the syndromic models and the spontaneous DNA mutations linked to the phenotype under study. In other words, this factor defines the similarity of the disease between the mouse and the human disorder in terms of the causal gene(s) as e.g., deducted from gene association and linkage analysis. As mentioned above, LCH is a human monogenic condition caused by a mutation in *RELN*. Therefore, the *Reeler* mouse fully meets the criterion of construct validity for the condition. There is also evidence for genetics to be associated with the etiology of several neuropsychiatric conditions, such as autism and schizophrenia, but, as the result of their multidimensional clinical symptoms, causal gene(s), if any, persist to be undiscovered [13]. Nonetheless, there are numerous genes associated with the human autistic pathology after analysis of Mendelian disorders (syndromes), rare mutations, or association studies; see e.g., [14].

Predictive validity, i.e., the similarity of the response to cures in humans and mice is difficult to establish, in the nonexistence of a recognized therapy in humans [14]. Thus, in the context of this discussion, face validity, i.e., the resemblance of the model phenotype to that of the human disorder, is the most important parameter to consider. 

Assessment of face validity in neuroscience translational studies requires a careful consideration of their behavioral and structural phenotypes. Broadly speaking, there are contradictory opinions as regarding the repetition in mouse of the human behavioral neuropsychiatric changes. This was, to some extent, predictable, as only a few trials, such as e.g., pre-pulse inhibition (PPI), which records sensory-motor responses, are highly comparable with only minimal modifications in the two species [15]. Notably, the issue has been the subject of several reviews on rodent models of autism, e.g., [16]. The conclusion of these surveys was that, although most of the models that have been used in drug discovery display behaviors with face validity for the human symptoms (i.e., deficits in social communication and restricted interests/repetitive behaviors), many drugs that were found to be useful in ameliorating these autism-related behaviors in mice were ineffective in humans.

Therefore, it becomes imperative to compare the structural alterations of the brains in the two species to substantiate or invalidate the models. We here summarize the state-of-art knowledge on the translational validity of homozygous (*reln*^−/−^) and heterozygous (*reln*^+/−^) *Reeler* mice with reference to the most common neuropsychiatric conditions directly or indirectly related to *RELN*. Because of its importance, we will primarily focus onto the brain structural modifications at magnetic resonance imaging (MRI) and histopathology in the two species.

## 2. The Reelin Gene and Protein

In humans, *RELN*, which has 94.2% homology with the mouse orthologue [8], is in chromosome 7q22 [17] and encodes for REELIN (RELN), a large glycoprotein of the extracellular matrix. The murine gene (*reln*) that also encodes for Reelin (Reln) was originally identified as the mutated gene in the *Reeler* mouse, which displays, among others, irregular lamination of the cerebral and cerebellar cortices, with an inversion of the regular ‘inside-out’ design observed in mammals [3,18]. The mouse and the human proteins have a similar size of 388 kDa. The structure of the protein recalls that of certain cell adhesion molecules, which specific cell types produce during brain and spinal cord development. 

In the neocortex, the Cajal–Retzius cells synthesize the glycoprotein and secrete it into the extracellular space [19]. Then, in post mitotic migrating neurons, Reln activates a specific signaling pathway that is required for proper positioning of these neurons. Northern blot hybridization showed that other areas of the fetal and postnatal brain also express the protein, with levels particularly high in cerebellum.

Reln is part of a signal transduction pathway that includes the apolipoprotein E2 (ApoER2), the very low-density lipoprotein receptors (VLDLR) and the cytoplasmic protein Dab1 [20]. Notably, the brain phenotype of mice with disruptions of *mDab1* or of both *apoER2* and *vldlr* closely resemble the brain of the *Reeler* mouse [21]. Another gene that interacts with the components of the Reln signaling pathways is platelet-activating factor acetyl hydrolase IB subunit α *(PAFAH1B1)*, also referred to as *LIS1* [22].

## 3. *RELN*-Related Human Neurological Conditions and Their Mouse Counterparts

Several human neurological conditions have a direct or indirect link with *RELN* and its encoded protein, as well as with the components of the RELN signaling pathway (Figure 1 and Table 1). We will briefly describe these conditions below, aiming to put in the better perspective those features that may be useful for well understanding the translational relevance of the *Reeler* mouse.

### 3.1. Neurological Conditions Caused by RELN Mutations

Several diseases are based on mutations of *RELN* or of genes encoding for proteins associated with the RELN signaling pathways. Among these, lissencephaly 2 (LIS2) and autosomal-dominant lateral temporal epilepsy (ADLTE) are of relevance to the present discussion as they have a clear genetic link with *RELN*.

#### 3.1.1. Human Lissencephalies and the Homozygous *Reeler* Mouse

Human lissencephalies are a group of cortical malformations that are consequent to neuronal migration disorders. Broadly speaking, the structural phenotype in lissencephalies ranges from a thickened cortex and complete absence of sulci (agyria) to a thickened cortex with a few, shallow sulci (pachygyria) [23]. The main feature of classic lissencephaly, formerly referred to as type I lissencephaly but today named lissencephaly 1 (LIS1), is a marked thickening of the cerebral cortex with a posterior to anterior grade of severity. An anomalous neuronal migration in the interval between the ninth to the thirteenth week of pregnancy causes LIS1, resulting in an assortment of agyria, mixed agyria/pachygyria, and pachygyria. An abnormally thick and ill ordered cortex with four highly disorganized layers, diffuse neuronal heterotopia, enlarged cerebral ventricles of anomalous shape, and, often, hypoplasia of the corpus callosum are typical of LIS1 [24]. The basal ganglia are normal, except that the anterior limb of the internal capsule is usually not noticeable, and, most often, the cerebellum is normal as well. 

Lissencephalies are now classified based on brain imaging results and molecular investigation [25], as they have been associated with mutations in several genes such as *LIS1* (*PAFAH1B1*; MIM#601545), *DCX* (Doublecortin; MIM#300121), *ARX* (Aristaless-related homeobox gene; MIM#300382), *RELN* (Reelin; MIM#600514), *VLDLR* (MIM#224050) and *TUBA1A* (αtubulin 1a) [26]. Some rare forms of lissencephaly (LCH) are associated with a disproportionately small cerebellum.

##### Lissencephaly 2

a) Humans

Lissencephaly 2 (LIS2) also referred to lissencephaly syndrome, Norman–Roberts type or Norman–Roberts syndrome (OMIM #257320) is associated with *LIS1* but displays several specific clinical features. In 2000, Hong and colleagues were the first to describe an autosomal recessive form of lissencephaly that, at MRI, also exhibited severe alterations of the cerebellum, hippocampus, and brainstem. More specifically, these alterations consisted of a thickening of the cerebral cortex with a simplified convolutional pattern that was particularly evident in the frontal and temporal lobes, whereas the parietal and occipital lobes were almost normal. The hippocampus was unfolded and flattened, lacking definable upper and lower blades. The corpus callosum was thin and the lateral ventricles enlarged. The cerebellum was clearly smaller than in the normal brain, hypoplastic, and devoid of folia. Authors also showed that the responsible gene mapped to chromosome 7q22 and that the condition was associated with two independent mutations in *RELN*, resulting in low or undetectable amounts of RELN after Western blots analysis of the patients’ serum [9]. Two other unrelated groups of patients, later, presented the same type of LIS2 [27]. They were children that, at MRI, displayed a 5–10 mm thick cerebral cortex, a malformed hippocampus and a very hypoplastic cerebellum, almost completely devoid of folia. As LIS2 is a rare disease, there are very limited histopathological data on the condition. To our knowledge, the only post-mortem description of a male fetus with Norman–Roberts syndrome reported the occurrence of a four-layered cerebral cortex (Figure 2A,B), a well-developed cerebellum with organized folia, and heterotopia of the dentate nucleus [28].

b) *Reeler* Homozygous Mice

Alterations in *Reeler* homozygous recessive mice fully recapitulate those in human LIS2 (Figure 1). Due to obvious technical and practical reasons, the amount of MRI data in mouse is by far less abundant than in patients, whereas mice have provided extensive histopathological information. The first MRI description of the neuroanatomical phenotypes in homozygous (and heterozygous) mice using morphometry and texture analysis, led to conclude that the structural features of the *Reeler* brain most closely copied the MRI phenotype of LIS2 patients [29]. Indeed, the *reln*^−/−^ mice had a smaller brain, but larger lateral ventricles compared to wild-type littermates. Sharp differences existed in the olfactory bulbs, dorsomedial frontal and parietal cortex, certain districts of the temporal and occipital lobes, and the ventral hippocampus where gadolinium-based active staining demonstrated a general disorganization with differences in the thickness of individual hippocampal layers. The cerebellum also resulted profoundly affected by the mutation and appeared strongly hypoplastic. A subsequent study, based on the use of manganese-enhanced MRI (MEMRI) to better detect the cortical laminar architecture, compared the MEMRI signal intensity in the cerebral cortex of normal and mutant mice. The authors of this survey observed that signal was low in cortical layer 1, increased in layer 2, decreased in layer 3 until mid-layer 4, and increased again, peaking in layer 5, before decreasing through layer 6. In *Reeler* there were, instead, no appreciable changes in signal intensity, an observation consistent with the absence of cortical lamination after histological examination [30]. A more recent and very elegant study has employed diffusion tractography imaging (DTI) to perform an in vivo origin-to-ending reconstruction of the mouse somatosensory thalamo-cortical connections and demonstrated an extensive remodeling in *Reeler* mutants because of the highly disorganized cortical lamination [31].

In keeping with the results of imaging studies, at gross anatomical examination the *reln*^−/−^ mouse brain was atrophic, as total volume in mutants decreased of about 19% when compared to normal mice [29]. Such a reduction was particularly evident in the cerebellum (Figure 2E,F) that also displayed a very limited degree of foliation. Therefore, also the gross anatomy of the *Reeler* brain closely resembled that of the LIS2 human brain. 

In general, it seems that the histological anomalies in mutants depended on an abnormal migration of neurons, rather than an alteration in cell fate determination or axonal guidance. Among these anomalies, the most distinguishing ones are that the cerebral and cerebellar cortices lose their layered structure, in accordance with the MEMRI observations [30]; numerous neuronal nuclei disappeared or, at least, became hardly recognizable in several brain regions; and neurons often displayed an ectopic position. Table 2 summarizes the most important structural anomalies of the *reln*^−/−^ CNS without taking into consideration the histological alterations in the cerebral cortex, hippocampus and cerebellum, as we will analytically discuss the phenotype of these brain areas in the following sections. Detailed descriptions of the morphological phenotype of the *Reeler* mouse CNS can be found e.g., in [29,32].

Very early observations demonstrated the occurrence of dendritic anomalies in cortical and hippocampal neurons of *Reeler* mice [48,49]. The discovery of *Reln* confirmed the dendritic pathology, as not only Reln but also the molecules of its signaling pathway resulted to be necessary for the correct maturation and differentiation of the dendritic branches and spines in hippocampal and neocortical pyramidal neurons [50,51].

Due to the complexity of the phenomena involved in dendritic maturation, one can argue that dendritic anomalies represented a consequence of the deep cytoarchitectonic derangement occurring in *Reeler* mice rather than a primary effect of the lack of Reln, but observations on heterozygous mice were not supportive of this interpretation [52,53]. Interestingly, the block of the Reln signaling by means of specific antibodies resulted in an increased complexity of branching in the apical dendrites of layer 2/3 cortical pyramidal neurons, whereas their basal arborizations remained unaffected [54].

There are many important issues related to the structure and role of the dendritic tree of neocortical and hippocampal pyramidal neurons that make the *Reeler* mouse an important tool for the study of (forebrain) neurodevelopment. Inputs to layer 5 neurons are processed by separate compartments, with the basal dendrites receiving bottom-up information and the apical dendrite being the recipient of a feedback input from higher cortical areas, see e.g., [55]. This framework is, however, even more complex because apical dendrites span most cortical layers before reaching layer 1, where the apical tuft is located [56]. Today we know well that the type and distribution of ion channels at the neurolemma ultimately determine the electrophysiological properties of a neuron. 

Essential to the function of the long apical dendrite of the pyramidal neurons is the progressively increasing density of hyperpolarization-activated cyclic nucleotide–gated (HCN) channels, proceeding from proximal to distal segments [57]. Such a gradient critically contributes to the functional distinction between dendritic compartments. Although Reln signaling specifies this gradient [58], 17β-estradiol, which stimulates Reln expression, promoted the enrichment of HCN1 in the distal dendritic compartment of CA1 neurons without the intervention of Reln [59]. The evidence that Reln was involved in the trafficking and targeting of ion channels in cortical and hippocampal neurons suggested that their intrinsic electrophysiological properties could indeed be different in the *Reeler* mouse. However, an early study by Bliss and Chung [60] demonstrated that, despite the layering derangement, the basic synaptic organization of the hippocampus was largely unchanged in mutants.

More recently, Silva et al. [61] carried out an accurate survey dealing with the intrinsic electrophysiological properties of cortical neurons in *Reeler* mice. These authors showed that the firing pattern and synaptic responses of the pyramidal neurons were normal, but with an inverted radial distribution. Notably, these authors concluded that, although mispositioned, neurons maintained the membrane properties appropriate to their function.

The apparent discrepancy between the data demonstrating the role of Reln in the modulation of ion channels and the relative lack of anomalies in the intrinsic properties of cortical neurons in mutant mice might have several explanations. Other factors, such as neuronal activity [62] could be more effective than Reln for the modulation of membrane channel targeting. Furthermore, the complex machinery of the long apical dendrite is required when layer 5 neurons settle appropriately but might be useless for the same neurons displaced to more superficial cortical layers. Finally, future investigations based on refined electrophysiological techniques, such as direct dendritic recordings, will help to establish if indeed the cortical neurons in *Reeler* mice display subtler changes of their firing/intrinsic properties that those so far ascertained.

Reln signaling is also able to modulate key molecules of the cascade leading to synaptic plasticity, such as the NMDA receptors [63,64]. Therefore, several studies concentrated on the changes of synaptic plasticity in *Reeler* mutants. Ishida et al. [65] reported that the induction of long-term potentiation (LTP) was impaired in the CA1 region of the hippocampus, claiming that the malpositioning of some neuronal populations could account for such an alteration. On the other hand, both the overexpression of Reln in transgenic mice [66] and Reln supplementation strongly increased LTP [67]. Later, a defective LTP was observed in the hippocampus of *vldr*-deficient mice, but slice perfusion with Reln was able to enhance LTP in CA1 [68].

Most cortical neurons are spiny, glutamatergic pyramidal cells, whose migratory path during prenatal development follows an inside-out radial pattern from the ventricular zone to the final position [69]. Reln signaling is essential for the localization of pyramidal neurons to appropriate cortical layers, as reviewed in [70]. Consequently, the lack of Reln caused a disruption of the layered cortical organization, including abnormal positioning [71,72], as well as an increased percentage of inverted pyramidal cells [73,74] (Figure 2C,D).

Inhibitory GABAergic interneurons represent a minority population within the neocortex. Yet, their morphological, neurochemical and functional diversity likely plays a key role for the cortical function, see e.g., [75]. Unlike pyramidal neurons, interneurons originate in the ganglionic eminence of the ventral telencephalon and follow a tangential migratory path to the cortex [69]. While the malpositioning of the pyramidal neurons in *Reeler* mice is evident, it is not clear if the Reln signaling cascade also affects the migration of the interneurons. An answer to this latter issue came from observations on *Reeler* mutants crossed with mice expressing the green fluorescent protein (GFP) in inhibitory neurons. Thus, the results of these observations confirmed that also the cortical interneurons displayed an abnormal laminar position and morphology [76]. However, we still do not know whether interneurons’ ectopy directly depends from Reln signaling or is rather the consequence of the malpositioning of principal pyramidal projection neurons. The debate on this issue is still open, as contradictory views exist in the literature. Namely, while some observations [77,78] argue against a direct role of Reln, Hammond et al. [79] showed that only early-generated cortical interneurons were misplaced as a consequence of the ectopy of the pyramidal neurons, whereas the correct layering of late-generated interneurons seemed to be directly modulated by Reln signaling.

Other basic neurodevelopmental features, such as cortical [80] and cerebellar neurogenesis, seem to be as well regulated by the glycoprotein. Consequently, the minicolumnar organization of the cerebral neocortex appeared to be deeply affected by Reln deficiency [81] and some physiological counterparts of cortical connectivity, such as trans-synaptic signal propagation, were also impaired [82]. However, the outcome of Reln deficiency on the microcircuitry sustaining the cortical machinery is controversial and, surprisingly, the deep architectonic disorganization that follows the lack of the protein occurs in the absence of dramatic functional anomalies. Both early and more recent studies point out that the absence of Reln did not prevent the development of functionally appropriate cortical connections and maps [31,83,84,85,86]. In addition, when studied at the fine-scale electron microscopic level, the basic synaptic organization of misplaced cortical neurons was unchanged [87]. Therefore, although the laminar organization is thought to be critical for cortical computation [88,89], evidences obtained in *Reeler* mice led Guy and Staiger [90] to challenge the importance of cortical lamination, affirming that “future studies directed toward understanding cortical functions should rather focus on circuits specified by functional cell type composition than mere laminar location”.

Macroscopically, the cerebellum of the *Reeler* mouse is smaller than that of age-matched littermates (Figure 2E,F); it is club-shaped with the main axis transverse to the mid plan of the body, and has an almost completely smooth surface, with just a few superficial grooves [91]. The architecture of the *Reeler* cerebellum is profoundly different from the normal pattern, firstly because of the impairment in the complicated series of migrations made by neurons to reach their destination in the mature organ. Trajectories of migrating neurons follow two opposite directions from the surface to the depth of the cerebellum and the other way around, depending from the species, the type(s) of neurons and the developmental stages (for details see e.g., [92]). Eventually, disturbances in the migration of the cerebellar neurons make that *Reeler* mice display a cerebellum that retains several features of immaturity.

The area of the cerebellar cortex in mutants was analyzed quantitatively during postnatal (P0–P25) development and resulted to be reduced compared to age-matched controls [93]. Reduction in the extension of the cortex was particularly evident in the molecular layer and the (internal) granular layer. Physiologically, as the cerebellum matured, the molecular layer became more and more populated by the parallel fibers, but, at P25, its increase in size was about one third in the mutants compared to *reln*^+/+^ mice [93]. Post-migratory granule cells, which are born in the temporary subpial external granular layer, progressively populate the (internal) granular layer during normal cerebellar development. This process was disturbed in *Reeler*, to the extent that, from P0 to P10, the granular layer of *reln*^+/+^ mice increased about five-folds in size, but only 2.6-fold in *reln*^−/−^, where it was significantly reduced in size to 0.62-fold that of normal mice after P10 [93]. In a different way from the cortex, the medullary body was larger in the mutants than in wild-type mice. Its progressively increasing area mainly reflected the ongoing myelination of the axons of the Purkinje neurons that abandon the cortex moving across the white matter to reach the cerebellar nuclei, as well as the expansion of the incoming afferent and departing efferent fiber systems. The mass of the medullary body augmented in relation to postnatal age irrespectively of the lack of Reln (*reln*^−/−^ 2.59, *reln*^+/+^ 1.93-fold), but, at P25, *Reeler* mice had a larger medullary body than normal mice (1.88-fold) [93]. In brief, *Reeler* mice had a reduced cerebellar cortex but a bigger medullary body than their *reln*^+/+^ littermates. The cerebellar hypoplasia was thus a consequence of a reduction in cortical magnitude and cellularity and the latter, in turn, resulted to be associated to measurable differences in the degree of cell proliferation and apoptosis, as well as imbalances in the timing of postnatal cortical maturation [93]. The same study led to conclude that density of proliferating cells was the most significant predictive factor to determine the cortical cellularity in *Reeler* [93]. Therefore, beside the well-defined consequences onto neuronal migration, the lack of Reln also caused a calculable deficit in neuronal expansion. Ultrastructurally, the cerebellar neurons underwent several different forms of programmed cell death during postnatal development and the deficit of Reln affected the kind and grade of neuronal death [94].

Perhaps the most striking histological feature in mutants is the lack of alignment of the Purkinje neurons to form a discrete intermediate layer in the cerebellar cortex (Figure 2G,H). Thus, in *Reeler*, only about 5% of the Purkinje neurons were in a normal position, 10% were still inside the cortex but in the granular layer, and the remaining 85% formed an internal cellular mass intermixed with the white matter [95,96,97]. Ultrastructurally, in *Reeler* there was a reduction in the density of the contacts between the Purkinje neurons and the parallel and climbing fibers, from P5 onward [98]. Functionally, both the normally placed Purkinje neurons and those ectopically dislocated in the granular layer displayed a 0–1 response to stimulation, indicating that, as in normal mice, they received a synaptic contact by a single climbing fiber. The Purkinje neurons in the internal cellular mass, instead, showed intensity-graded responses to electrical stimulation, as several climbing fibers provided them with a convergent input [95], likely as a failure of physiological pruning to occur [99]. Neurochemically, there were no obvious variations between normal mice and the mutants in the temporal expression of some widely diffused neuronal and glial markers (NeuN, vimentin, calbindin, GFAP, Smi32, GAD67) during postnatal development [93], but the Bergmann glia was misplaced in *Reeler* [100].

To conclude, the histological and electrophysiological observations in *Reeler* mutants suggest that similar structural and functional alterations may also occur in LIS2 patients, particularly in relation to the postnatal growth retardation, severe intellectual disability, and spasticity observed in affected subjects (see also https://www.orpha.net/).

##### Lissencephaly 3

*TUBA1A* mutations [101,102] cause lissencephaly 3 (LIS3), another human condition that has a mouse counterpart. TUBA1A chiefly occurs in cortical, hippocampal, cerebellar and brainstem post-mitotic neurons, with expression falling soon after birth but persisting through adulthood [103]. The mouse phenotype consists, among others, in a failure of the cerebellar Purkinje neurons to migrate, so that, similarly to *Reeler*, they remain entrapped into the medullary body, where they form a series of streaks intermingled with the neurons of the cerebellar nuclei [104]. Several other mutations of *TUBA1A* exist in humans. They give rise to a predominant phenotype of LCH, which also shows irregularities of the corpus callosum and the basal ganglia/internal capsule [105].

#### 3.1.2. Autosomal-Dominant Lateral Temporal Epilepsy and the Heterozygous *Reeler* Mouse

ADLTE, also referred to as autosomal dominant epilepsy with auditory features, partial epilepsy with auditory aura or partial epilepsy with auditory features, is a genetic epileptic syndrome, clinically showing typical focal seizures in response to specific sounds. ADLTE is genetically heterogeneous, and mutations in the leucine-rich, glioma inactivated 1 gene (*LGI1*) account for fewer than 50% of affected families. Very recent observations demonstrated that heterozygous *RELN* mutations give rise to a classic ADLTE syndrome, clinically identical to that associated with mutations of *LGI1*. Seven different heterozygous missense mutations in *RELN* were, in fact, described in some unrelated families of Italian ancestry with familial temporal lobe epilepsy-7 (ETL7–OMIM #616436) [106]. Incidence was 17.5% over the total number of families studied that were specifically suffering by lateral temporal lobe epilepsy [106]. By three-dimensional modeling, the same authors anticipated that the outcomes of these mutations would be protein structural defects and misfolding. Some of the affected individuals displayed a reduction up to 50% of their serum levels of the 310 kDa RELN isoform in comparison to healthy subjects and thus, very likely, the mutations also resulted in a loss of function. In a subsequent study on the same patients, 1.5 T MRI scans were not useful in detecting structural anomalies of the brain [107]. Similarly, in a very recent study on an 18-year old ADLTE patient, 3T MRI brain scans could not provide relevant information on indistinct grey-white matter connections, voxel-based morphometry, and cortical thickness [108]. However, analysis of functional connectivity with high-density electroencephalography (HdEEG) revealed greater local synchrony in the left temporal (middle temporal gyrus), left frontal (supplementary motor area, superior frontal gyrus), and left parietal (gyrus angularis, gyrus supramarginalis) regions of the cerebral cortex and the cingulate cortex (middle cingulate gyrus) as compared to normal subjects [108]. 

As the discovery of RELN mutations in ADLTE is a quite recent finding, there are, at present, no observations on heterozygous *Reeler* mice focused to ascertain possible similarities with the human phenotype. Like ADLTE patients, *reln*^+/−^ mice display a 50% reduction of Reln in their serum. Therefore, it would be interesting to investigate whether sound-triggered epileptic manifestations also occur in these animals. A very recent study has demonstrated that optogenetic stimulation of the parvalbumin (PV) immunoreactive GABAergic neurons of the mouse basal forebrain can modulate the cortical topography of auditory steady-state responses [109]. As the regional distribution of these neurons displayed relevant differences in *reln*^+/−^ mice compared to wild-type animals [110,111], any phenotypic alteration may be of interest to shed additional light onto human ADLTE. Finally, a very latest report has provided proof of concept that HdEEG can be used to record electrical activity from the mouse brain in a model of juvenile myoclonic epilepsy [112]. Therefore, one can envisage applying such an approach to analyze the brain electrical pattern in *reln*^+/−^ mice aiming to collect data for translational comparison with ADLTE.

### 3.2. Human Conditions Caused by Mutations of Genes of the Reln Intracellular Pathway and Their Mouse Correlates

In general, the brain phenotype of the human monogenetic conditions that are consequent to mutations of the genes coding for the proteins of the RELN intracellular signaling pathway is similar to that of the *reln*^−/−^ mouse brain, except that, in certain cases, differently from *Reeler*, the human cerebellum is normal at MRI and gross anatomical observation (Figure 1). We will briefly describe these conditions below.

#### 3.2.1. *VLDLR*-Associated Cerebellar Hypoplasia

*VLDLR*-associated cerebellar hypoplasia is an autosomal recessive genetic form of non-progressive congenital ataxia [113]. The main clinical symptom of the condition is a predominantly truncal ataxia with retarded ambulation, so that children either learn to walk after six years of age or never walk without aid. Dysarthria, strabismus, moderate-to-profound intellectual disability, and seizures are other features of the disorder. MRI findings comprise hypoplasia of the inferior portion of the cerebellum, affecting both the vermis and the hemispheres; pachygyria of the cerebral hemispheres with a negligibly but uniformly thickened cortex in the absence of a neat anteroposterior gradient, reduction is size of the brainstem, particularly the pons. The condition is monogenic, and due to mutations in *VLDLR*.

*Vldlr* only knock-out mice did not show the drastic brain phenotype that can be seen in double knock-out mice devoid of *vldlr* and *apoER2*, which, instead, recapitulate in full the phenotypic alterations of *Reeler* mutants or *dab1* knock-out mice [21,114]. As Reln interacts with both Vldlr and ApoER2, clear functional differences in how these two receptors transduce the glycoprotein signal have been postulated [114]. That the interaction of Reln with Vldrl occurs with much lower affinity than with ApoER2 [115] could explain the less severe phenotype of the *vldlr* knockout mice compared to *Reeler*. Remarkably, alterations that in mouse followed the knocking-out of *vldlr* were particularly noticeable in cerebellum and consisted in failure of the Purkinje neurons to form a well-defined monolayer and reduction of their dendritic arbor [114]. They thus recall in full the human MRI phenotype of *VLDLR*-associated cerebellar hypoplasia.

#### 3.2.2. Spinocerebellar Ataxia Type 37

Spinocerebellar ataxia type 37 (SCA37) is a late onset syndrome that affects adults, with dysarthria, slowly progressive gait and limb ataxia, severe dysmetria in the lower extremities, mild dysmetria in the upper extremities, dysphagia, and abnormal ocular movements. In most cases, the first clinical signs encompass tumbles, dysarthria, or stiffness followed by a typical cerebellar syndrome. The early presence of altered vertical eye movements is a characteristic clinical feature of SCA37 that foregoes the symptoms of ataxia. The progression is slow and affected individuals usually become wheelchair bound between ten and thirty-three years after the onset of the disease [116]. At MRI, there is an initial atrophy of the vermis. Later, atrophy rapidly affects the entire cerebellum, without alterations of the brainstem [117]. Molecular analysis has shown that an unstable repeat insertion in *DAB1* is the cause of the cerebellar degeneration and, on the basis of the genetic and phenotypic evidence, the mutation has been proposed as the molecular basis for SCA37 [118].

Notably, the *dab1* deficient mice that derived from a spontaneous mutation called *Scrambler* or from gene knockout were phenotypically indistinguishable from the homozygous *Reeler* mice [119].

#### 3.2.3. PAFAH1B1-Associated Lissencephaly/Subcortical Band Heterotopia

*PAFAH1B1*-associated lissencephaly/subcortical band heterotopia, also referred to as *LIS1*-associated lissencephaly/subcortical band heterotopia, encompasses Miller–Dieker syndrome (MDS), isolated lissencephaly sequence (ILS) and, infrequently, subcortical band heterotopia (SBH) [120]. MRI findings for lissencephaly are the absence or the abnormal broadening of cerebral gyri, and the aberrant thickness of the cerebral cortex. Less frequently, it may be possible to observe an enlargement of the lateral ventricles, mild hypoplasia of the corpus callosum and of the cerebellar vermis. In *PAFAH1B1*-associated SBH, just beneath the cortex of the parietal and occipital lobes there are subcortical bands of heterotopic gray matter separated from the superficial cerebral cortex by a thin layer of white matter. Histologically, the cerebral cortex in LIS1-associated lissencephaly consists of four layers: a poorly defined marginal zone, which, however, has a very high cell density; a superficial neuronal layer with diffusely scattered neurons; a deeper neuronal layer with relatively sparse neurons; and a deepest neuronal layer with neurons arranged in columns.

The architectural alterations of the human cerebral cortex and hippocampus can be somewhat recapitulated in genetically engineered mice. For example, the overexpression of *pafah1b1* disturbed neuronal migration and layer formation in the developing cerebral cortex [121], whereas *lis1* deficiency in homozygous mice resulted in early embryonic death and in heterozygous mice led to a derangement of the normal hippocampal organization with ectopy of the granule cells [122]. Of note, Lis1, the protein encoded by *Pafah1b*, is part of the Pafah1b complex and binds, downstream of the Vldlr receptor, to Dab1 that becomes phosphorylated in response to Reln [123].

### 3.3. Human Conditions Possibly Related to RELN Mutations and Their Mouse Correlates

#### 3.3.1. Spinocerebellar Ataxia Type 7

Spinocerebellar ataxia type 7 (SCA7) is an autosomal-dominant neurodegenerative syndrome that outcomes from polyglutamine expansion of ataxin 7 (ATXN7). Remarkably, although ATXN7 has a widespread expression in SCA7 patients, the pathology primarily hits the cerebellum and the retina [124]. A recently published paper suggested that RELN might be a formerly unidentified factor accountable for the tissue specificity of SCA7 [125].

#### 3.3.2. Autism and the Heterozygous Reeler Mouse

The disorders of the autistic spectrum (ASD), which are characterized by social, behavioral, and language insufficiencies, comprise Asperger syndrome, autism, and pervasive developmental disorder-not otherwise specified (PDD-NOS). Less than 20% of these disorders, acknowledged as ‘‘syndromic autism’’, derives from monogenetic diseases, most commonly fragile X syndrome and tuberous sclerosis. The remaining 80% of ASD cases are classified as ‘‘non-syndromic autism’’ and are widely investigated to find candidate genes that may contribute to pathology [126].

##### Genetics

At present autism cannot be considered, strictly speaking, a genetic disease, as one or more causative gene(s) has (have) not been found yet. The first gene association study implicating *RELN* in autism dates to 2001 [127]. However, subsequent gene population surveys yielded contrasting results [128,129,130,131]. Nonetheless, a more recent meta-analysis showed that at least one single nucleotide polymorphism (SNP) in *RELN* could be significantly associated with the risk of autism [132]. Therefore, results of SNP analysis appear to be compatible with the idea that heterozygous mutations in *RELN* may contribute to the onset of the disorder. Genetic studies on autism led to two main outcomes: 1. the more predominant existence of rare or de novo inherited mutations of a number of genes in autistic patients; 2. the discovery of certain common gene variants that contribute to the risk of autism but are also present, albeit at lower frequency, in the normal population [133]. When more than two de novo mutations occur in a gene, the latter becomes a very likely causative candidate of a disorder. There are four unique documented de novo mutations of *RELN* associated with autism [134,135,136], thus implicating *RELN* as a possible cause of ASD. However, although nonsense mutations are more frequent in autistic patients than in controls after whole-exome sequencing, there is not a striking gross increase of de novo mutations in the former [135]. To study autism heritability, one can also employ a different approach that distinguishes total narrow-sense heritability from that due to common gene variants. By this method, it was concluded that narrow-sense heritability of autism is ∼52.4%, and that the main contribution heritability was due to common gene variants, whereas rare de novo mutations contribute only for about 2.6% of cases, but substantially influence individual liability to the condition [137]. Thus, *RELN* may primarily have a role in the individual *predisposition* to manifest autism rather than being one of the contributory causes of the disorder.

Further support for a RELN involvement in autism derived from the detection of reduced expression of the *RELN* transcript and protein in autistic individuals. Decreased RELN levels were apparent in the superior frontal cortex [10] and cerebellum of autistics as compared to controls [10,11,138]. In these areas, the levels of *RELN* mRNA were lower, as was the *DAB1* transcript, whereas *VLDLR* mRNA levels augmented.

##### Imaging

Imaging findings in autism have been recently reviewed [139]. Numerous observations join to prove that there is an atypical development of the brain in autistic children. Early cross-sectional studies demonstrated that the brain of these children had a higher volume than that of regularly developing subjects. However, growth curves in the two groups eventually met at later childhood. More specifically, in the 6–35-year interval, there was an initial period of brain overgrowth, and then growth slowed down or even stopped during early and late infancy to which a phase of fast reduction of the brain volume eventually followed [140]. Neuroimaging data also indicated that differences in the brain of autistic people started to be detectable within the first two years after birth, *before* clinical symptoms became obvious. There are conflicting views about the probability that an accelerated growth rate of the brain in this postnatal window goes together with the occurrence of early neurodevelopmental perturbations [139]. In relation to this, it is relevant that we still do not know when the initial neuropathological signs of autism occur, also from the paucity of studies on autistic children during the first year of life.

The mechanisms at the basis of the abnormal growth of the autistic brain are also unclear. Although most imaging studies have focused onto the gray matter of the cerebral cortex, there are data indicating that an increased amount of cerebrospinal fluid in the subarachnoid space [141] and/or a greater volume of the white matter [142] occurred in parallel to the enlargement of the autistic brain. As regarding the cerebral cortex, its surface, but not thickness, increased in the autistic brain [143]. 

To summarize, that an initial brain overgrowth may be a reliable biomarker for autism remains highly questionable. Thus, it seems more profitable to focus onto regional brain structural differences in a more effective search for new neuroanatomical findings of clinical relevance [139].

Before entering the description of regional MRI investigations in autism, it is important to stress that, at present, there are no specific and/or causative objective findings for the condition, but, instead, the very same regions altered in autism may be interested in other psychiatric conditions.

The individual constituents of the neural circuitries causal of ASD are well defined. They include regions of the fronto-temporal, fronto-parietal, and dorsolateral prefrontal cortex (PFC); parts of the limbic system; the fronto-striatal circuitry, and the cerebellum. Neuroimaging studies on these regions have employed different approaches such as the definition of a region-of-interest (ROI), voxel- or vertex-vise methods. Traditional ROI studies have reported atypical findings in brain areas that participate to social cognition such as the medial PFC, the anterior cingulate cortex (ACC), the inferior frontal cortex, the superior temporal sulcus, the amygdala, and the anterior insula.

The cerebellum was larger than in controls in several MRI studies on autistic patients older than three years [144], but not in younger children [143]. Differently from the cerebellum, the size of the vermis was smaller [145,146,147] or larger [146] or did not display any relevant difference [147], and such discrepancies possibly depend from the different clinical presentations of the condition [147,148]. It is also unclear whether there are differences in size of individual vermal lobules, as claimed by some authors [145], but not others [147]. Similarly, there were no differences between cerebellar hemispheres in one study [147], whereas another group has found the hemispheric size as the only significant structural dissimilarity between verbal and nonverbal subjects [149].

The still fuzzy picture emerging from the imaging studies onto the autistic brain makes it very difficult to compare the human and mouse data in the search for common biomarkers. To our knowledge, there are only two MRI studies on the brain of the heterozygous *Reeler* mouse. In the first, Badea and co-worker [29] reported that the total volume of the brain, the ventricular volume and the hippocampal volume correspondingly raised of about 6%, 82%, and 7% compared to normal control mice. However, after statistical analysis, they showed that these volumes were like those of *reln^+/+^* normal mice. They also measured the areas of different parts of the brain in comparison with wild-type mice and found no differences in hippocampus and cerebellum, but an enlargement of the lateral ventricles. A more recent paper confirmed the ventricular enlargement, but found a reduction of the cerebellar volume, whereas the volume of the motor cortex as well as its thickness was unchanged [150]. Therefore, given the paucity of data in mouse and the still unclear MRI pattern in human autistic subjects, one can only conclude that, at present, the cerebellum could be a part of the brain deserving further imaging investigations for translational purposes.

##### Histopathology

A series of histological alterations of the whole brain occur in the autistic brain. In the first histological surveys, the only cortical area showing qualitative structural abnormalities was the ACC that, in autistic patients, lacked architectural refinement and had only a coarse lamination [151]. However, in the following decades substantial amounts of data have been collected and the list of cerebral structures displaying histopathological changes in ASD has grown substantially to include a series of cortical regions, the amygdala, the cerebellum and the brainstem, see e.g., [152,153]. Below we will briefly summarize the most significant histopathological findings in human patients and compare them to those in the heterozygous *Reeler* mouse. However, the interpretation of both human and mouse finding needs often much caution, because not all studies used sound quantitative approaches and/or proper stereological procedures.

a) Changes Affecting the Whole Brain

The diffuse alterations observed in the brains of autistic subjects at postmortem included cortical dysplasia and neuronal heterotopia, with the formation of aggregates of neuronal cell bodies in anomalous positions [154]. Other alterations, i.e., differences in size of the neuronal nucleus and perikaryon, occurred at the cytological level. These differences started being evident in young children and became more apparent in adults, but then tended to re-equilibrate with time [155]. Remarkably, there might be some compensations between different areas, as in some parts of the brain neurons were bigger, but smaller in others.

In the autistic brain there was also an increase of the neuropil extension in certain but not all cortical areas that have been investigated so far [156]. It is unclear which neuropil component(s) is (are) responsible of these volumetric variations. Fewer dendrites were, in fact, immunostained for microtubule-associated protein 2 (MAP2) in the PFC [157] and a reduction of dendritic spines was reported in hippocampus [158], but other studies reached completely opposite conclusions after examination of the pyramidal neurons from layers 2 and 5 in the frontal, temporal, and parietal cortex [159]. The issue of dendrite and dendritic spines density is quite important in the general framework of this discussion, because these parameters have been widely investigated, primarily aiming to validate the heterozygous *Reeler* mouse as a translational model.

Alterations in neuronal differentiation and migration may also occur in the autistic nervous system and, thus, the consequences of a dysregulation of these processes may be at the basis of whole brain changes in autism [154]. In spite of this, there are only a few investigations on the expression of RELN in the brain of autistic patients and, after quantitative analysis, there was no alteration in the density of layer 1 RELN+ neurons in the superior temporal lobe of the autistic brain, although these neurons represent about 70% of the total layer 1 population [160].

b) Brain Regional Changes

Forebrain

Most of the histopathological observations on the brain of autistic patients focused on the cerebral cortex and hippocampus and, broadly speaking, alterations were almost exclusively restricted to neurons. The parameters considered have been size, number, and density of the different neuronal populations, often in relation to the cortical layers or the hippocampal subfields. A point of attention in considering these studies is that, in several cases, comparative brain volume evaluations between autistic patients and controls are missing, while they are, instead, essential to settle whether modifications in cell density reflect true differences in total cell counts.

Cerebral Cortex

The autistic pathology affects several regions of the cerebral cortex. 

The PFC, which plays a major role in cognitive control, displayed a general overgrowth with an increase in the number of neurons, whereas glial cells were apparently unaffected. Among the GABAergic interneurons, there was a numerical increase of the PV-immunoreactive chandelier neurons, whereas the calbindin- and calretinin-expressing neurons were unaltered [153,161,162,163]. However, after qRT-PCR, the levels of the RELN and GAD67 mRNAs diminished in the post-mortem PFC from autistic patients in comparison to healthy controls [164]. 

In the inferior frontal cortex, changes affected the small-sized pyramidal neurons that did not display numerical alterations but were of smaller size [163].

The fusiform gyrus, which intervenes in facial recognition and social interactions, had a reduced neuronal density in layer 3, whereas neurons in layers 2, 5, and 6 were less numerous, being also smaller in layers 5 and 6 [165], but these alterations were not confirmed in [166]. 

In the frontoinsular cortex and ACC, both intervening in emotional regulation and self and others awareness, lamination was rudimental. In the former, von Economo neurons (VENs) of layer 5 increased in number [167,168,169], whereas in ACC neuronal density augmented in layers 1–2 of area 24a of the left hemisphere but diminished in layers 5–6 of area 24c; size, instead, diminished in all layers of area 24b [151,170].

The anterior midcingulate cortex also displayed a numerical increase of VENs, as well as of the pyramidal neurons of layer 5 that, however, were of smaller size [171]. 

Lastly, the entorhinal cortex, which has a role in memory, navigation and perception of time, displayed characteristic terminal swellings, referred to as spheroids [172] that were also observed in all hippocampal subfields (see below).

Another issue of interest is the possibility that there are alterations in the minicolumnar organization of the cerebral cortex in early age onset autism [156], as it might be the case in *Reeler* mice. Specifically, it appeared that minicolumns were smaller, more numerous and with lower neuronal density in several cortical areas of autistic (and Asperger’s syndrome) patients, although these observations still need to be confirmed in full [156]. Under this perspective, it may be useful to here recall some of the results on the localization of the neural cell adhesion molecule 2 (NCAM2) in the *Reeler* mouse because the molecule has also been proposed as a predisposition gene for the development of autism [173]. In mutants, NCAM2 immunopositive and negative patches formed a mosaic filled with dendritic aggregates originating from two different populations of neurons in a fashion suggestive of a minicolumnar organization [174,175]. However, one must consider these findings with much caution, as that minicolumns are indeed the fundamental modular units of neocortical organization is currently still a matter of debate, see e.g., [176] for review.

Whereas the human cerebral cortex has been widely investigated in autism, investigations on the cerebral cortex of the heterozygous *Reeler* mouse have been relatively few but reported a reduction in the levels of GAD67 [53,177] like that observed in humans. Since the studies on hippocampus led to highly comparable results in the two species, it would be of importance to undertake rigorous investigations on the number and density of the cortical neurons in *reln*^+/−^ mice, with attention to the different neurochemical populations of inhibitory interneurons. The results of these studies will be relevant, from one side, to validate the mouse model and, from the other, to confirm some numerical observations in humans that, as mentioned at beginning of this section, need validation using approaches more reliable than those often employed at histopathology.

Hippocampus and Amygdala

In humans, beside the widespread occurrence of spheroids, other general changes in hippocampus [151,158,172,178,179] consisted in a reduction of neuronal size and dendritic arbors, and these smaller neurons appeared to be more densely packed. There were also a series of specific modifications affecting the excitatory pyramidal neurons that were more numerous in CA1, but less abundant in all other adjacent hippocampal regions. The GABAergic inhibitory interneurons, instead, displayed a higher density, specifically the calbindin-immunoreactive neurons in the dentate gyrus, the parvalbumin-immunoreactive neurons in CA1 and CA3, and the calretinin-immunoreactive neurons in CA1 [151,158,172,178,179].

In the heterozygous *Reeler* mouse the hippocampus displayed reduced levels of GAD67, [53,177] that could be somewhat restored after stereotaxic injections of Reln [67,180]. These experiments indicated the existence of a causal link between the decrease in GAD67 expression and Reln haplodeficiency. In keeping with such a possibility, Reln supplementation could, at least partly, reverse such a decrease. Other experiments, in line with this interpretation, have confirmed that the decrease in the levels of GAD67 in heterozygous mice can be overturned, e.g., after administration of nicotine, which reduces the GAD67 promoter methylation and increases its transcription [177].

In heterozygous mice, pyramidal neurons displayed a reduction in the average length and width of their apical and basal dendritic spines [52], consistent with the decrease in the spread of the dendritic arbor of the same population of neurons in humans. Additionally, Reln supplementation was effective in promoting a full (apical) or partial (basal shaft) spine recovery [180]. These morphological observations are in line with a previous report showing that, in the forebrain, spines were hypertrophic in mice conditionally overexpressing Reln [66]. At electrophysiological recordings CA1 pyramidal neurons displayed reduced spontaneous inhibitory postsynaptic currents [181], an observation that was fully coherent with the reduction of the inhibitory input from the GABAergic interneurons observed histologically.

Synaptic plasticity is fundamental for hippocampal function. In CA1 of heterozygous mice, LTP was impaired [182] as well as long-term depression (LTD) [181], which returned to normal levels after the administration of Reln [180]. Additionally, in *reln*^+/−^ and *reln*^−/−^ mice, post tetanic potentiation (PTP), a form of short-term plasticity that depends on neurotransmitter release, was reduced in CA1 [180,183] and could be reversed by Reln [180].

Collectively these data indicate that the experimental administration of the glycoprotein was able to reverse the morphological, neurochemical, and physiological hippocampal deficits consequent to a reduction of brain Reln. Translationally, they are very important because they offer some cues for further investigations onto the autistic brain. It would be of interest to map the post-mortem distribution of hippocampal RELN in patients compared to healthy controls, to ascertain whether the pattern of immunoreactivity will be consistent between humans and mice.

In the autistic patients, several studies reported that the amygdala, which is involved in emotional learning, increased in size and displayed an augmented density of neurons within the medial, central, and cortical nuclei [151,184,185,186,187]. Neurons were, however, less numerous, although numerical variations could be age dependent. To our knowledge, Boyle et al have investigated in detail the amygdala of the homozygous *Reeler* mouse with a marker-based phenotypic approach [188], but there are no data on heterozygous mice.

Cerebellum

Analysis of cerebellar alterations in autism has attracted many efforts of the basic researchers and clinicians. The most consistent anatomic findings in autistic patients were a reduction in size of certain lobules of the cerebellar vermis (but see 3.3.2b Imaging) and a decrease in the number [186,189,190,191,192] and size [193,194] of the Purkinje neurons. The inhibitory GABAergic basket and stellate interneurons that connect with these neurons did not show quantitative differences compared to normal cerebella [195]. This observation is indicative of a late developmental death of the Purkinje neurons, as they differentiate well before the interneurons. In addition to structural observations, Western blots demonstrated a reduction of about 40% in the level of expression of RELN in autistic patients related to age and sex corresponding controls [11]. Quantitative RT-PCR also showed a drop of RELN and GAD67 mRNAs in the post-mortem autistic brain [164].

Notably many of the histological alterations in the human autistic cerebellum are like those described in *reln*^+/−^ mice. These animals displayed a progressive loss of Purkinje neurons already during the first weeks of life [35], and inferior numbers of these cells were observed in adult subjects as well [196]. Human MRI studies did not allow, at present, to ascertain whether the cerebellar vermis is hit by the pathology in its entirety or, rather, only at specific lobuli. Therefore, our group has, at first, focused its attention on five different lobules, which receive diverse types of afferent functional inputs, to analyze the number and topological organization of the Purkinje neurons in *reln*^+/+^ and *reln*^+/−^ adult mice of both sexes [197]. We have thus shown that the Purkinje neurons: 1. Displayed a lower density in *reln*^+/−^ males (14.37%) and *reln*^+/−^ females (17.73%) compared to *reln*^+/+^ males; 2. Were larger in *reln*^+/−^ males than in the other phenotypes under study, and smaller in females (regardless of the *reln* genetic background) than in *reln*^+/+^ males; 3. Were more messily arranged along the YZ axis of the vermis in *reln*^+/−^ males than in *reln*^+/+^ males and, except in central lobule, *reln*^+/−^ females.

Very recently, as many observations have associated a number of synapse-related genes in the genesis of autism and other neuropsychiatric conditions [198,199], we have examined the expression of synaptophysin 1 (SYP1) and contactin 6 (CNTN6) in the vermis of *reln*^+*/*−^ and *reln*^+*/*+^ adult mice of both genders [200].

SYP1 is a pre-synaptic marker and CNTN6 is a marker of the synapses made by the parallel fibers onto the Purkinje neurons’ dendrites. Notably, there is evidence, although still to be validated in full, that SYP1 is involved in the structural alterations of the autistic synapses [189,201], and very recent observations have shown that copy number variations [202] or a truncating variant [203] of *CNTN6* are found in autistic patients. In addition, *CNTN6* mutations may be a risk factor for several neurodevelopmental and neuropsychiatric disorders [150,199,204].

In line with these human studies, we have demonstrated that *reln*^+*/*−^ mouse males displayed a statistically significant drop of 11.89% in SYP1 compared to sex-matched normal animals, whereas no modifications were detected comparing *reln*^+*/*+^ and *reln*^+*/*−^ females [200]. In *reln*^+/−^ male and female mice, reductions in SYP1 levels were particularly evident in the molecular layer, whereas in heterozygous mice of both sexes a reduction in CNTN6 occurred in all the three cortical layers of the vermis. In addition, alterations in the levels of expression of SYP1 in the molecular layer of male *reln*^+/−^ mice ensued across all lobules except lobule VII, but they were limited to lobule II for the granular layer and lobule VII for the Purkinje cell layer.

Thus, the widespread reduction of SYP1 and of CNTN6 in the molecular layer of *reln*^+*/*−^ male mice well matched with the autistic phenotype in humans [150].

In the vermis (and the whole cerebellum), there is proof for a topographic segregation of the areas controlling motion versus those connected to cognitive and affective functions, and the diverse lobules are coupled with precise zones of the brain and spinal cord [205]. The CNS areas that handle sensorimotor inputs are directly or indirectly connected with the anterior lobe (lobules I–V of the vermis), lobule VIII, and, to a lesser grade, with lobule VI; on the contrary, cortical association areas that collect non-motor responses are linked to lobules VI and VII. Existing clinical data indicate that the vermis is the chief target of the limbic system, and physiological and behavioral observations implicate the vermis in the regulation of emotions [206]. Therefore, the neurochemical modifications of the cerebellar cortex in heterozygous mice are fully in line with the possibility that the social and communication aberrations typical of autism rest on anomalies of the limbic system and its connections [207,208].

At post-mortem, a numerical reduction of the Purkinje neurons in the posterior cerebellum was long ago described in autistic subjects [184,209], but it did not appear to disturb the vermis [189]. Hypoplasia in lobules VI and VII was initially detected in vivo using MRI [145], but subsequent observations proved the existence of two distinct autistic subtypes related to vermian hypoplasia or hyperplasia [146]. A systematic review and meta-analysis of the accounts of structural MRI has then established that the reduction in size of lobules VI–X (i.e., the lobules included in the posterior cerebellum) showed a remarkable heterogeneity that associated to differences in time of life and intelligence quotient (IQ) merely in lobules VI–VII [210]. Other observations showed that the posterior/inferior vermis, i.e., lobules VII, VIIIb (left), and IX, was more prone to pathological deviations [211], with a decrease of the gray matter after quantitative MRI [190,212,213]. Therefore, it appears that the cerebellar phenotype of the heterozygous *Reeler* mouse is fully compatible with that in humans and that a deeper structural and neurochemical characterization may be useful to direct the discovery of new biomarkers of translational interest.

#### 3.3.3. Schizophrenia and the Heterozygous *Reeler* Mouse

Schizophrenia is a ruinous psychiatric condition that affects about 1% of the population. Its main clinical symptoms are hallucinations, delusions, and cognitive disturbances. These symptoms derive from brain dysfunctions that derive from genetic and environmental factors [214]. However, schizophrenia is not strictly a genetic disease, although gene deletions, duplications, and variations may be risk factors for the disorder. At present, the gene(s) that could be involved in the pathology remain elusive for the most (see OMIM #181500), but a microdeletion in a region of chromosome 22, called 22q11, was recently established to be involved in a small percentage of cases [215].

Genetic studies have shown a link between *RELN* and schizophrenia [216] and, over the past ten years, many SNPs in the *RELN* gene loci occurred in parallel with the beginning and/or severity of the clinical signs [217], but results still are under debate and need further verification [218]. One should perhaps emphasize that observations on gene expression have converged to show that the genes implicated in schizophrenia are more highly expressed during fetal than postnatal life [219], thus making more difficult to ascertain their true role in the etiology of the condition.

##### Imaging

Structural MRI findings in schizophrenia have been recently reviewed [220]. There is enough information to propose that the condition is associated with a continuing development of gray matter aberrations, chiefly throughout the first stages of the disease. Reduction of the depth of the cerebral cortex in the superior temporal and inferior frontal regions was reported in individuals that later became psychotic. In patients with first episode psychosis, there was, instead, a reduction in the thickness of the superior and inferior frontal cortex, and in the volume of the thalamus. In chronic schizophrenia, the gray matter decreased further in the frontal and temporal areas, cingulate cortices, and thalamus, particularly in patients with unfortunate outcomes. Structural modifications of the white matter occurred only in a small number of longitudinal studies.

As the human phenotype is still very far from clear, it is not surprising that the few MRI observations in *Reeler* mice are still insufficient to draw any definitive consideration of translational relevance (for MRI data on Reeler see 3.3.2 Imaging).

##### Histopathology

Although gross structural alterations of the brain were lacking, subtle pathological changes in specific populations of neurons and in cell-to-cell communication occurred in schizophrenic patients, see [221] for a recent review. Histopathology mainly consisted in modifications of the number and density of neurons at the level of the whole brain and/or specific neuronal subpopulations, and in morphological and neurochemical alterations of these neurons.

a) Cerebral Cortex

As discussed above for autism, the most widely investigated area of the brain has been the PFC that, in general terms, displayed an increased neuronal density and an altered neuroplasticity with age-related differences between normal and schizophrenic subjects. More specifically, a statistical meta-analysis of thirty papers published between 1993 and 2012, concluded that the density of cortical neurons increased with age irrespective of the condition, but the rate of accretion was much slower in the schizophrenics [222]. However, other cortical areas, such as e.g., the dorsal ACC, displayed no changes in neuronal and glial densities after stereological analysis [223].

The above-mentioned meta-analysis [222] has also taken into consideration the density of inhibitory neurons after immunolabeling with GAD67, PV, or calbindin, and found that it was greater in schizophrenic patients compared to controls before the age of 40, but lower thereafter.

Notably, both Reln and GAD67 mRNAs were downregulated in the PFC of schizophrenic subjects with no relation to neuronal damage [224]. In keeping with these observations, it appeared that in the PFC there was a vulnerability of the inhibitory circuits, with markers of the inhibitory interneurons showing some of the more consistent alterations [225]. More precisely, these alterations consisted in a reduction in the levels of the GAD67 mRNA and protein in subsets of GABAergic basket cells containing PV [222,226,227] or cholecystokinin (CCK) [226]. Notably, these two populations of basket cells are responsible of the inhibition of the pyramidal neurons giving rise, respectively, to the cortical θ and γ oscillations altered in schizophrenia. In addition, the pyramidal neurons targeted by the PV+ basket cells expressed lower levels of the GABA_A_ receptor α1 [226].

It is also interesting that the levels of RELN and GAD67 mRNAs in microdissected GABAergic neurons of PFC layer 1 were lower in schizophrenics, but unchanged in layer 5 of the same patients [228]. In addition, in the dorsolateral division of the PFC, the GABAergic chandelier neurons targeting the axon initial segment of the pyramidal neurons displayed remarkable neurochemical alterations. These changes were particularly evident in layers 2/3, where immunoreactivity for the GABA membrane transporter GAT1 diminished, in parallel with an increase of the GABA_A_ receptor α2 subunit in the axons of their target pyramidal neurons [229].

Occurrence of dendritic spine pathology was another prominent feature of the schizophrenic human brain [230,231]. Spine loss mainly affected the smaller spines of the pyramidal neurons in layer 3 of the neocortex and arose during development, possibly because of altered mechanisms of generation, pruning, and/or upkeep [230].

As already mentioned in the section dedicated to autism, investigations on the cerebral cortex of the heterozygous *Reeler* reported a reduction in the levels of GAD67 [53,177], in full accordance with the human studies. A study carried out on mice whose mothers were stressed during pregnancy showed that the downregulation of Reln and GAD67 was associated with a hypermethylation of their promoters [232], this being one the mechanisms in support for the contribution of an altered epigenetic control in the down-regulation of RELN expression in schizophrenia, see [233] for review.

b) Hippocampus

The human hippocampal pathology in schizophrenia is by far less clear than in the cortex. Some initial studies have, in fact, reported a decrease in area or overall volume of the hippocampus, or in the number, size, and density of neurons, as well as a disarray of the pyramidal cells, with greatest differences affecting the pyramidal cell density in left CA4; however, several other subsequent surveys were negative, see [234] for review. In any case, hippocampal alterations in schizophrenics are not specific, as they display several common traits with those in autism.

We have previously discussed the histological changes in the hippocampus of the heterozygous *Reeler* mice in relation to autism. These modifications recall, in toto or in part, those in schizophrenia. Additional information detailed, in individual hippocampal layers, the decrease of neuronal GAD67 in CA1, CA2 and dentate gyrus, and the reduction of PV immunoreactive interneurons in CA1 and CA2, in the perspective to validate these mice as a model of schizophrenia [111]. In translational terms, the aforementioned impairment of LTP in heterozygous mice [182] is of interest, as it also occurs in schizophrenic patients [235].

c) Cerebellum

In the cerebellum of schizophrenic patients, there was a loss of distal and terminal dendritic branches and a decrease in density of the dendritic spines of the Purkinje neurons [236]. Again, these histological alterations are not specific for the condition, being evident also in autism. In addition to such changes, in the schizophrenic cerebellum there were altered levels of expression of the general presynaptic marker SYP1, of complexin II, a marker of the excitatory synapses, but not of complexin I that, instead, labels the inhibitory synapses [237]. Thus, some of the structural changes described in the heterozygous *Reeler* mouse in relation to autism also in cerebellum recollected the human schizophrenic phenotype.

## 4. Does the Behavior of Heterozygous *Reeler* Mice Recall the Human Conditions Related to RELN?

The recapitulation of the behavioral modifications typical of human autism, schizophrenia, or epilepsy in heterozygous *Reeler* mice still is a subject of debate. The dissimilar outcome of behavioral experiments performed in different laboratories is not surprising, because neuropsychiatric behaviors in humans primarily regard social interaction, communication, and restricted interest, and these behaviors are, obviously, very difficult to measure objectively in mice [238].

It is perhaps worth mentioning here that most of our knowledge on the effects of Reln in the cognitive or behavioral field derives from work on mouse hippocampus. This is not surprising as this part of the brain, as discussed previously, has been the primary focus of numerous investigations also in human patients affected by neuropsychiatric disturbs. Several behaviors comparable to those observed in these human disturbs also occur in *reln*^+/−^ mice [239,240,241], as well as the deficits in reversal learning after visual discrimination tasks that were hypothesized to follow a diminished visual attention [240]. In addition, testing *reln*^+/−^ mice for anxiety-related behavior, motor impulsivity and morphine-induced analgesia yielded a different behavioral profile from that of wild-type littermates in that they displayed, starting form adolescence, a decreased inhibition and emotionality. To these modifications, a small increase of impulsive behavior and different pain thresholds also occurred in adult mice [242]. Heterozygous mice were also tested in a complex series of PPI protocols (unimodal and cross-modal) to conclude that they exhibited a multifaceted configuration of changes in startle reactivity and sensorimotor gating, with both resemblances to and dissimilarities from schizophrenia [243]. At least partly in line with these latter observations, other studies failed, incompletely or in full, to validate the behavioral analogies between neuropsychiatric patients and *reln*^+/−^ mice [181,244,245,246,247,248]. For example, Salinger and co-workers were unsuccessful to find differences between *reln*^+/−^ and *reln*^+/+^ mice after testing gait, emotionality, social aggression, spatial working memory, novel-object detection, fear conditioning, and sensorimotor reflex modulation [244]. In another survey [246], heterozygous *Reeler* mice were evaluated for cognitive plasticity in an instrumental reversal learning task, impulsivity in an inhibitory control task, attentional function in a three-choice serial reaction time task, and working memory in a delayed matching-to-position task to conclude that there were no differences in comparison to *reln*^+/+^ littermates in prefrontal-related cognitive trials. However, *reln*^+/−^ mice were deficient in two operant tasks. From these observations, the authors concluded that heterozygous *Reeler* mice were *not* a good model for the essential prefrontal-dependent cognitive shortfalls detected in schizophrenia, although they could be useful to model learning deficits in a more general sense.

In another paper it was reported that heterozygous and wild-type mice displayed comparable levels of general activity, coordination, thermal nociception, startle responses, anxiety-like behavior, shock threshold; identical cued freezing behavior, and comparable spatial learning in Morris water maze tasks, albeit a significant decrease in contextual fear conditioned learning was observed in *reln*^+/−^ mice only [181]. These authors have then hypothesized that the pharmacological administration of Reln in heterozygous mice could restore the response to PPI. They were unable to find differences in the acoustic startle reflex among treated and untreated animals, but Reln-treated *reln*^+/−^ mice showed a substantial increase in the percent inhibition to 78-, 86- and 90-dB pre-pulse [180].

One study has specifically focused onto the *reln*^+/−^ mouse behavioral phenotype in young (P50–70) and fully adult (older than P75) animals to conclude that they were not useful to model schizophrenia [245]. An ample series of behavioral test was used (Irwin test; rotarod; spontaneous locomotor activity; social behavior; light-dark transition; startle response and pre-pulse inhibition; hot plate). Heterozygous mice were like their wild-type littermates at either age, though completely adult male *reln*^+/−^ mice were involved in social exploration for a longer time. In addition, performance on the rotarod deteriorated with age.

Indeed, age appeared to be a further issue of complexity. In fact, adult *reln*^+/−^ mice did not display discernible changes in activity, motor coordination, anxiety, or environmental perception compared to wild-type littermate controls. However, juvenile animals displayed not as much of anxiety- and risk assessment-related behaviors in the elevated plus-maze [182,241]. In addition, in one of these two studies it was demonstrated that young *reln*^+/−^ mice had a hippocampal-dependent shortfall in associative learning and impulsivity–anxiety-related behavior [182]. Additionally, one study, starting from the clinical observations that reported the occurrence of vocal and motor anomalies in autistic patients, has described that *reln*^+/−^ mice had a general delay in the development of their repertoire of neonatal vocal and motor behaviors [249].

Finally, one must consider that gender apparently influenced some behaviors, although very few studies have focused on this issue. Among these studies, young heterozygous female mice were described to be more active in the light/dark transition test than the heterozygous males that were, instead, more aggressive than females during social interaction [241].

## 5. Usefulness of the *Reeler* Mouse in Translational Studies: Concluding Remarks

The analysis of the literature discussed above requires one trying to draw some conclusions about the true usefulness of the *Reeler* mouse in translational studies.

At first, it may perhaps be useful to remember that, as discussed, *RELN* is causative of LIS2 and a small percentage of ADLTE, whereas only tentative associations up to now hold for the other conditions here considered (see Figure 1 and Table 1).

Remarkably, both LIS2 and ADLTE are rare diseases. Very few cases of LIS2 (around ten) so far come about in the literature (see OMIM #257320). Similarly, patients with lateral temporal epilepsy (LTE) are only about 10% of all temporal epilepsies, and the real prevalence of ADLTE, which has been up to now reported in Europe, USA, and Japan, is unknown, but it may account for about 19% of familial idiopathic focal epilepsies [250,251]. When one considers the human conditions related to the Reln signaling pathway, one still encounters a group of rare diseases. The actual frequency of *VLDLR*-associated cerebellar hypoplasia is unknown, but initial reports regarded not more than twenty-five affected individuals in Canada and USA [113,252], although the condition occurs worldwide, *PAFAH1B1*-associated lissencephaly is very rare as the prevalence of classic lissencephaly ranges from 11.7 to 40 per million births [120]. To date, sixty-six affected individuals and seven asymptomatic individuals with the ATTTC repeat insertion within *DAB1* have been reported in ten relatives from the south of the Iberian Peninsula, and no individuals with SCA37 from other geographic areas have been described [116]. SCA7 has a prevalence of less than 1:100,000 and accounts for about 2% of all SCAs [253].

Therefore, one must deal with the paradox of the relatively little interest for translational studies on the very conditions for which the *Reeler* mouse and/or mice with mutations of the genes of the Reln pathway fully meet the criteria for construct and face validity. Thus, the homozygous Reeler mice appear to be more interesting to the neurobiologist than to the clinician and their study will surely be still rewarding in terms of our comprehension of neurodevelopment, as the model already helped to establish that many functional and circuit features of cortical neurons are relatively independent from positional cues and cortical lamination, e.g., [179].

Very differently, the prevalence of autism in the worldwide population is around 1% [254] and that of schizophrenia is just below [221]. As the two conditions are very diffuse in the human population, there is an obvious translational interest for the heterozygous *Reeler* mouse as a model for the two disorders. However, such an interest is again paradoxical, as the validity of these mice to extract information about the human pathologies remains dubious. The first explanation for this uncertainty lies, beyond any doubt, in the substantial lack of construct validity, which is the direct consequence of the complex genetic background of autism and schizophrenia. As regarding face validity, the present survey of the literature clearly points out that there are several similarities but also dissimilarities between the human and the mouse phenotypes. Among dissimilarities, one must consider the heterogeneity of results of the behavioral experiments in mouse. Further complexity derives by the vast array of clinical symptoms in humans. Structurally, most of the imaging and post-mortem findings in humans are not specific for each of the two conditions. However, one must consider that both the human and mouse phenotype converges to indicate the cerebral cortex, hippocampus, and cerebellum as the primary foci of the pathologies and the inhibitory interneurons as major players in the context of the circuitry involved.

A serious drawback to a full validation of the heterozygous *Reeler* mouse as a model of autism and/or schizophrenia lies in the observation that the alterations so far described in mouse are very subtle in both structural, functional and neurochemical terms. The relatively low resolution of current neuroimaging procedures, and the difficulty to obtain post-mortem samples amenable for neurochemical, electrophysiological, and fine (ultra) structural analyses make it very difficult to establish whether the alterations in heterozygous *Reeler* mice have a true biological significance that goes beyond mere statistics [255,256]. In the affirmative, one could take advantage of these alterations to discover novel biomarkers that will be helpful for an earlier and more precise diagnosis in the human practice.

## Figures and Tables

**Figure 1 jcm-08-02088-f001:**
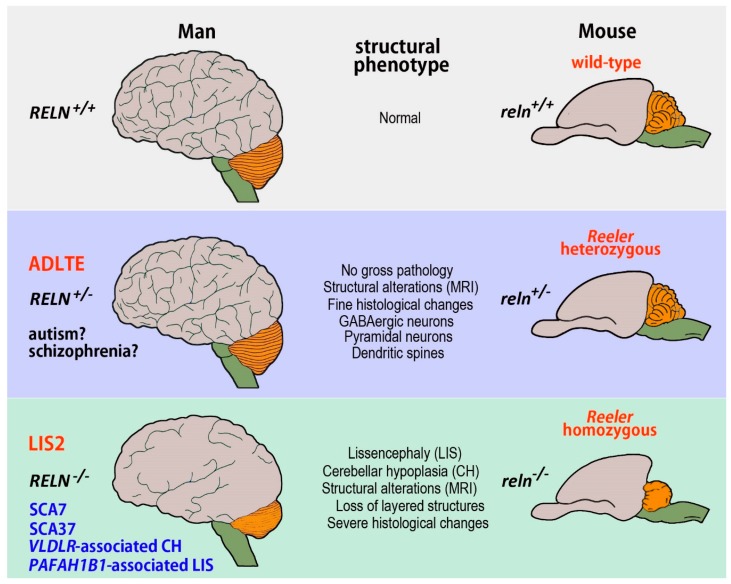
Summary of the most relevant human pathologies modeled in the *Reeler* mouse. The monogenic conditions provoked by the *RELN* gene, i.e., ADLTE and LIS2, are in red, those related to genes encoding for the proteins of the Reln intracellular cascade or only tentatively linked to RELN are indicated in blue. Autism and schizophrenia, which have a complex multifactorial etiology, are in black with an interrogative mark to underline the still tentative association of the two disorders with *RELN*. Abbreviations: ADLTE autosomal-dominant lateral temporal epilepsy, LIS2 lissencephaly 2, PAFAH1B1 platelet-activating factor acetyl hydrolase IB subunit α, *RELN* reelin gene (human), *reln* reelin gene (mouse), SCA37 spinocerebellar ataxia type 37, SCA7 spinocerebellar ataxia type 7, VLDLR very low-density lipoprotein receptor.

**Figure 2 jcm-08-02088-f002:**
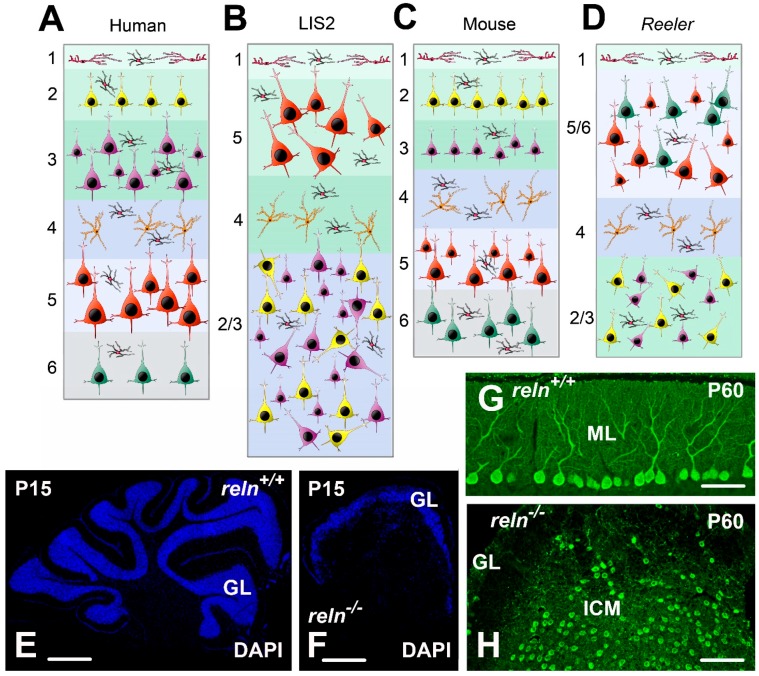
Structural alterations in human, LIS2, and homozygous *Reeler* mouse (**A**–**D**)**;** modifications of the neocortex architecture in human LIS2 (**B**); and *Reeler* mutation (**D**); compared to healthy controls (**A**,**C**). After MRI imaging, the human LIS2 cortex is thicker than normal, whereas there are apparently no thickness changes in mouse. Note that in both species the pathological neocortex only consists of four layers, with an upside-down layer disposition mainly affecting the pyramidal neurons that are also irregularly oriented compared to their usual positioning in normal individuals/mice. Pyramidal neurons are in different color and sizes according to their position in cortical layers. Stellate spiny cells of layer 4 are orange. Inhibitory interneurons are black with a red nucleus. Cajal-Retzius cells of layer 1 are red. (**E**–**H**): Structural alterations in the *Reeler* mouse cerebellum; (**E**) sagittal sections of the P15 cerebellum in a normal *reln^+/+^* mouse; and (**F**) a *Reeler reln^−/−^* mouse: the *Reeler* cerebellum is much smaller and devoid of folia, with a smooth surface. (**G**) Misalignment of the Purkinje neurons in the P60 cerebellum of the *Reeler* mouse. After calbindin 28 kDa immunostaining, the Purkinje neurons are well aligned in a monolayer below the molecular layer in *reln^+/+^* mice. They, instead, form a large internal cellular mass within the white matter in *reln*^−/−^ mutants (**H**). Abbreviations: DAPI = 4′,6-Diamidine-2′-phenylindole; GL = granular layer of the cerebellar cortex; ICM = internal cellular mass; ML = molecular layer of the cerebellar cortex; P = postnatal day.

**Table 1 jcm-08-02088-t001:** Summary list of the human neurological conditions related to the *RELN* gene.

Disease	Transmission	Causative Gene(s)	*Reeler* Mutants of Translational Interest	Other Mouse Models
LIS 2	Autosomal recessive	*RELN*	Homozygous	see text
ADLTE	Autosomal dominant	*RELN* (in 17.5% of cases)	Heterozygous	*LG11*-mutated
*VLDLR*-associated cerebellar hypoplasia	Autosomal recessive	*VLDLR*	Homozygous	*VLDLR* knock-out
SCA37	Autosomal dominant	*DAB1*	Homozygous	*DAB1* knock-out *apoER2* knock-out
*PAFAH1B1*-associated lissencephaly	Autosomal dominant	*PAFAH1B1*	Homozygous	*Lis1^+/−^*
SCA7	Autosomal dominant	*ATXN7*	Homozygous	SCA7 knock-in
Autism	Isolated cases Multifactorial	see https://omim.org # 209850	Heterozygous	see text
Schizophrenia	Autosomal dominant	see https://omim.org # 181500	Heterozygous	see text

Note that only LIS2 and autosomal-dominant lateral temporal epilepsy (ADLTE) have a demonstrated link with *RELN*. *RELN* may be relevant for *LIS1*.

**Table 2 jcm-08-02088-t002:** Main histopathological changes in the homozygous *Reeler* mouse.

Division of CNS	Region/Division	Subdivision/Nucleus	Type(s) of Alteration	Ref
**Forebrain**	Olfactory bulb		Slight disruption of the glomerular layer. Numerical reduction and clustering of granule cells	[33,34]
	Striatum		Decreased PV-immunoreactivity	[35]
	Diencephalon		Misrouting of GnRH neurons to the cerebral cortex	[36]
		Mammilary bodies	Alteration of projections to hippocampus	[37]
**Midbrain**	Rostral colliculus		Loss of individual limits in the three more superficial layers Spread of corticotectal projections Anomalies of retinotectal projections	[38]
	Mesencephalic nucleus of V		Spread of neurons along their route of migration	[39]
	Substantia nigra		Anomalous clustering lateral to the ventral tegmental area	[40]
	Medulla oblongata and pons	Dorsal cochlear nucleus	Partial loss of layered organization	[41]
		Inferior olivary nucleus	Loss of folding - Swelling	[42]
		Somatic motorneurons (Nucleus ambiguous, facial and trigeminal)	Slight displacement and loss of somatotopic organization (muscolotopy)	[6,43]
		Pontine nuclei	Ventral shift	[44]
**Spinal cord**	Dorsal horn (laminae I-II)	Nociceptive	Abnormal neuronal positioning	[45]
	Lateral horn	Preganglionic sympathetic and parasympathetic neurons	Abnormal neuronal positioning	[46,47]

The Table does not list the histopathological observations on cerebral cortex, hippocampus, and cerebellum.

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
