# Peer review of "The Reeler Mouse: A Translational Model of Human Neurological Conditions, or Simply a Good Tool for Better Understanding Neurodevelopment?"

_jcm, 2019, doi:10.3390/jcm8122088_

Round 1

Reviewer 1 Report

The review discuses whether Reeler mouse  can be considered as a model of autism, SZ, or other neuropsychiatric disorders. The possible role of reein gene mutation in Spinocerebral ataxia, Cerebellar aplasia, and autism are properly are analyzed satisfactory. However the role of reelin  deficiency in SZ is not properly discuss. For reference to the reelin down  regulation in GABAergic neurons of SZ patients the Authors should refer to Guidotti et al. Front.Cell Neurosci.19,89,2016. and to Grayson and Guidotti Neuropsycopharmacology 38, 1038, 2013.

Author Response

The review discusses whether Reeler mouse can be considered as a model of autism, SZ, or other neuropsychiatric disorders. The possible role of reelin gene mutation in Spinocerebral ataxia, Cerebellar aplasia, and autism are properly are analyzed satisfactory. However the role of reelin  deficiency in SZ is not properly discuss. For reference to the reelin down  regulation in GABAergic neurons of SZ patients the Authors should refer to Guidotti et al. Front.Cell Neurosci.19,89,2016. and to Grayson and Guidotti Neuropsycopharmacology 38, 1038, 2013.

We thank Reviewer 1 for her/his appraisal of our work. In reposnde we have expanded the manuscript section dealing with Schizophrenia (3.3.2) and referred to the above suggested references.

Reviewer 2 Report

The review ‘The Reeler mouse: a translational model of human neurological conditions or simply a good tool for better understanding neurodevelopment?’ focuses on diseases which have been genetically linked to Reelin or genes of proteins from the Reelin signaling cascade and discusses the value of reeler and heterozygous reeler mice to model these diseases. The positive aspect of this review is the very thorough and detailed description of phenotypes associated with the different diseases and those observed in reeler and heterozygous reeler mice.

The major drawback of this review to my opinion is its structure. It is difficult to read as it is mostly just a juxtaposition of lots of facts without getting the straight connection between human and mouse as suggested in the title. I think it would be better to include the reeler results directly in the part describing the human disease phenotype to point out similarities and differences at once and to follow the main issue of this review more straightly.

One additional suggestion would be a table stating the human disease, the different phenotypes associated with this disease and then how much this could be correlated to findings in the reeler mouse e.g. is it the same, is it unchanged or even the opposite effect? This could replace table 2+3.

In addition to this major issue I have also some minor suggestions and requests:

If table 2+3 should be maintained, they need some revision – they look unfinished. In order to have more space to state the specific alterations more clearly (which is the most important point here) I would suggest to converge ‘general histology’ and ‘specific neuronal alterations’ and include the ‘Refs’ directly at the corresponding statement. # as ‘Number’ should be added to the legend.

Figure 1 - the indication of 'structure' on the mouse cortex, even showing differences between reeler and wildtype, is misleading as there is of course a smooth cortical surface in all mice.

Line 404-405 – please specify this consistent alterations

Line 482 – here is a clear bias for one of the two cited studies. Although 128 contradicts 127 only 127 is further discussed.

Line 573 – specify what is meant with ‘both directions’?

Line 711 – ‘In line’ does not fit very well here as neither promotor methylation nor nicotine is mentioned before.

Line 733ff – the cerebellum is discussed in much detail – a table or figure would make sense to depict all these detailed effects in human disease compared to reeler.

Figure 2 does not add important information – it would be easy to just list the cells which are affected – in this case also the literature could be stated in the overview. The Figure would be of great value, if it would show the morphological differences comparing 'healthy' – 'disease' – 'reeler', but this is not the case in the current form.

One additional more general issue would be to include the DNA-methylation subject when discussing Reelin and schizophrenia. The genetic link might be weak but DNA-methylation leads to a decrease in Reelin mRNA and protein levels in about 50% of schizophrenia patients regarding to Impagnatiello et al. 1998 – PNAS 95:15718-15723.

Author Response

The review ‘The Reeler mouse: a translational model of human neurological conditions or simply a good tool for better understanding neurodevelopment?’ focuses on diseases which have been genetically linked to Reelin or genes of proteins from the Reelin signaling cascade and discusses the value of reeler and heterozygous reeler mice to model these diseases. The positive aspect of this review is the very thorough and detailed description of phenotypes associated with the different diseases and those observed in reeler and heterozygous reeler mice.

We thank very much Reviewer 2 for her/his appreciacion of our work.

The major drawback of this review to my opinion is its structure. It is difficult to read as it is mostly just a juxtaposition of lots of facts without getting the straight connection between human and mouse as suggested in the title. I think it would be better to include the reeler results directly in the part describing the human disease phenotype to point out similarities and differences at once and to follow the main issue of this review more straightly.

We thank the reviewer for this very useful suggestion. In reponse we have discussed in paraller the human and mouse phenotype and pointed out similarities and differences of the two for each of the human conditions described in the paper.

One additional suggestion would be a table stating the human disease, the different phenotypes associated with this disease and then how much this could be correlated to findings in the reeler mouse e.g. is it the same, is it unchanged or even the opposite effect? This could replace table 2+3.

Again thank you very much for this suggestion. Since we have discussed the differences between the human and mouse phenotypes in response to the previous comment, and it would have been almost impossible to summarize these differences in tabular for, we have deleted Table 2 in the original manuscript and described the human phenotype in autism in text. Original Table 3 has been maintained, a little improved, and is now Table 2.  

In addition to this major issue I have also some minor suggestions and requests:

If table 2+3 should be maintained, they need some revision – they look unfinished. In order to have more space to state the specific alterations more clearly (which is the most important point here) I would suggest to converge ‘general histology’ and ‘specific neuronal alterations’ and include the ‘Refs’ directly at the corresponding statement. # as ‘Number’ should be added to the legend.

See respone to comment above.

Figure 1 - the indication of 'structure' on the mouse cortex, even showing differences between reeler and wildtype, is misleading as there is of course a smooth cortical surface in all mice.

Figure 1 has been amended accordingly to show the same cortical smooth surface in all mouse genotypes.

Line 404-405 – please specify this consistent alterations.

We have added a brief description of the most relevant alterations (now at lines 797-799)

Line 482 – here is a clear bias for one of the two cited studies. Although 128 contradicts 127 only 127 is further discussed.

We have briefly discussed also ref. 128 (now ref 59 at lines 234-235).

Line 573 – specify what is meant with ‘both directions’?

We have rephrased the sentence as follows:  Trajectories of migrating neurons follow two opposite both directions from the surface to the depth of the cerebellum and the other way around…(now at lines 305-307).

Line 711 – ‘In line’ does not fit very well here as neither promotor methylation nor nicotine is mentioned before.

We have rephrased this sentence as follows: Other experiments, in line with this interpretation, have confirmed that the decrease in the levels of GAD67 in heterozygous mice can be overturned, e.g. after administration of nicotine, which reduces the GAD67 promoter methylation and increases its transcription [176]. - now at lines 652-654.

Line 733ff – the cerebellum is discussed in much detail – a table or figure would make sense to depict all these detailed effects in human disease compared to reeler.

We have added images of the cerebellum in the new figure 2.

Figure 2 does not add important information – it would be easy to just list the cells which are affected – in this case also the literature could be stated in the overview. The Figure would be of great value, if it would show the morphological differences comparing 'healthy' – 'disease' – 'reeler', but this is not the case in the current form.

We have substituted Figure 2 with a new one in which we have depicted the main changes in the cerebral cortex in LIS2 and Reeler, as well as some of the most striking modifications of the Reeler cerebellum.

One additional more general issue would be to include the DNA-methylation subject when discussing Reelin and schizophrenia. The genetic link might be weak but DNA-methylation leads to a decrease in Reelin mRNA and protein levels in about 50% of schizophrenia patients regarding to Impagnatiello et al. 1998 – PNAS 95:15718-15723.

We have introduced the issue above at lines 817-820: A study carried out on mice whose mothers were stressed during pregnancy showed that the downregulation of Reln and GAD67 was associated with a hypermethylation of their promoters [232] , this being one the mechanisms in support for the contribution of an altered epigenetic control in the down-regulation of RELN expression in schizophrenia, see [233] for review.

We have quoted 233 Guidotti A., Grayson D. R., Caruncho H. J. Epigenetic RELN dysfunction in schizophrenia and related neuropsychiatric disorders. Front Cell Neurosci 2016, 10, 89 and not Impagnatiello et al 1998 because we found no direct mention of DNA methylation in that paper.

Round 2

Reviewer 2 Report

The review ‘The Reeler mouse: a translational model of human neurological conditions or simply a good tool for better understanding neurodevelopment?’ improved a lot in its revised form. I have only some minor additional corrections:

Line 122-124 Check the wording of the sentence. ‘Among the diseases based on mutations of RELN ….” might be an appropriate correction to this sentence

Table 1 row 2 last columns – the text does not fit into this column – perhaps you can just shift this comment to column 1 or place it as an annotation at the end of the table?

There are several mistakes in the index numbers (e.g. line 186, 492) In addition I would suggest to simplify the index.

 Author Response

Thank you for the very careful revision of the MS.

We have corrected all issues accordingly.